# Mobile LiDAR Scanning System Combined with Canopy Morphology Extracting Methods for Tree Crown Parameters Evaluation in Orchards

**DOI:** 10.3390/s21020339

**Published:** 2021-01-06

**Authors:** Kai Wang, Jun Zhou, Wenhai Zhang, Baohua Zhang

**Affiliations:** 1College of Engineering, Nanjing Agricultural University, Nanjing 210031, China; wk65010@stu.njau.edu.cn (K.W.); wen_hai123@126.com (W.Z.); bhzhang@njau.edu.cn (B.Z.); 2Jiangsu Province Key Laboratory of Intelligent Agricultural Equipment, Nanjing 210031, China

**Keywords:** canopy measurement, mobile scanning system, lightweight state estimator, tracked robots

## Abstract

To meet the demand for canopy morphological parameter measurements in orchards, a mobile scanning system is designed based on the 3D Simultaneous Localization and Mapping (SLAM) algorithm. The system uses a lightweight LiDAR-Inertial Measurement Unit (LiDAR-IMU) state estimator and a rotation-constrained optimization algorithm to reconstruct a point cloud map of the orchard. Then, Statistical Outlier Removal (SOR) filtering and European clustering algorithms are used to segment the orchard point cloud from which the ground information has been separated, and the k-nearest neighbour (KNN) search algorithm is used to restore the filtered point cloud. Finally, the height of the fruit trees and the volume of the canopy are obtained by the point cloud statistical method and the 3D alpha-shape algorithm. To verify the algorithm, tracked robots equipped with LIDAR and an IMU are used in a standardized orchard. Experiments show that the system in this paper can reconstruct the orchard point cloud environment with high accuracy and can obtain the point cloud information of all fruit trees in the orchard environment. The accuracy of point cloud-based segmentation of fruit trees in the orchard is 95.4%. The R^2^ and Root Mean Square Error (RMSE) values of crown height are 0.93682 and 0.04337, respectively, and the corresponding values of canopy volume are 0.8406 and 1.5738, respectively. In summary, this system achieves a good evaluation result of orchard crown information and has important application value in the intelligent measurement of fruit trees.

## 1. Introduction

The fruit tree canopy is the main focus of photosynthesis and fruit production, and canopy information can reflect the yield and growth status of fruit trees and can provide a certain level of guidance for fertilization, irrigation, pruning, and other operations. Consequently, the development of a high-precision and efficient canopy morphology measurement method has certain practical significance. Traditional measurement methods include tape measurements and projections [1]. These tasks are time-consuming and laborious, and the measurement accuracy is closely related to worker proficiency. Ground LiDAR has also been applied to canopy measurements and has achieved a good level of accuracy [2]. However, it is a tedious method, and its high cost hinders its promotion and development. In recent years, with the development of sensors and SLAM technologies [3], mobile LiDAR scanning systems have been increasingly used in areas such as robot environment perception and autonomous driving, which have laid a solid foundation for the use of mobile LiDAR scanning systems in fruit tree canopy measurements.

At present, cameras and LiDAR are the most commonly used instruments for measuring tree canopy parameters, but cameras are consumer products and have an advantage over LiDAR in terms of price. Ding et al. [4] used a digital camera to estimate the canopy volume of pear trees with the help of least squares and five-point parameter calibration. Dong et al. [5] reconstructed a 3D model of a canopy using an RGB-D camera and estimated the morphological parameters using a semantics-based mapping algorithm. Sun et al. [6] used an Unmanned Aerial Vehicle (UAV) imaging system to create point cloud maps of orchards and to determine canopy information on fruit trees. LiDAR has advantages over camera equipment in terms of measurement distance and range accuracy as well as environmental adaptability, which have led to more widespread application of LiDAR in the field of tree canopy measurement. Previous studies have relied heavily on the fusion of Global Navigation Satellite System (GNSS) and LiDAR to perform canopy parameter measurements. Martínez Casasnovas et al. [7] used vehicle-mounted 2D LiDAR and GNSS to scan olive orchards and to estimate olive tree canopies, and James P et al. [8] used GPS and LiDAR to construct an orchard map and database of canopy information for precise orchard management. André Freitas Colaço et al. [9] developed a mobile ground-based laser scanner suitable for large commercial orange groves to estimate canopy volume and height, relying on GNSS with LiDAR and using the alpha-shape method to perform canopy measurements. It can be seen from the above that GNSS is widely used in orchard mapping, but due to excessive canopy height and density in orchards, GNSS can suffer from signal loss and canopy measurement failure [10]. To address this situation, morphological measurement technology for fruit trees based on laser SLAM has been developed in recent years. Zhou et al. [11] used an Mobile LiDAR Scanning (MLS) system based on the LiDAR Odometry and Mapping (LOAM) algorithm to measure tree diameters at breast height. Pierzchała et al. [12] used a wheeled robot with 3D LiDAR to map a forest using graph-SLAM technology and to extract the diameter at breast height of a single tree. The above methods all used the LiDAR-IMU loose coupling method for laser mapping. When the environment has fewer feature points, it is easy to encounter mismatching, map confusion, and other scenes [13]. The recently developed LiDAR-IMU tightly coupled odometry algorithm has greatly improved the accuracy and robustness of point cloud map construction. The application of this technology enables better results for tree crown measurements.The Tightly Coupled 3D Lidar Inertial Odometry and Mapping (LIO-mapping) algorithm [14] uses the Visual-Inertial State Estimator (VINS-mono) monocular odometry’s dynamic initialization scheme [15], which relies on variable motion with roll angles, but large agricultural tracked robots cannot perform movement in rolls and pitches in a garden or orchard. The Robocentric lidar_inertial state Estimator (R-LINS) algorithm [16] uses a static solution to complete the system initialization and uses the method based on iterative error state Kalman filter (ESKF) to perform state estimation.

According to the needs of morphological parameter orchard measurements, we designed a crawler-type agricultural robot as a test platform for research and testing in orchards. The first step combines the LiDAR and IMU sensors carried by the robot to create a 3D point cloud map model. Then, a joint processing method based on filtering and European clustering was used to segment the fruit tree, and the k-nearest neighbour (KNN) search algorithm was used to recover the canopy point cloud information. Finally, some important fruit tree morphology parameters, such as tree height and crown volume, were extracted. This method can provide theoretical support and technical guidance for canopy measurements and can provide accurate decision information for orchard managers.

## 2. Materials and Methods

### 2.1. Experimental Platform

The mobile platform used in this study was a self-developed electric-driven agricultural tracked robot (as shown in Figure 1). The robot was equipped with a 20-kW lithium battery module, which ensured the robot’s long-term battery life in the orchard. An Advantech MIC-7700 industrial computer was used as the on-board control unit. This industrial computer has rich Input/Output (IO) interfaces, and good anti-vibration and anti-interference performances, and the operating system used was Ubuntu16.04, with an i7-5820k processor, 32 G memory, and 256 G solid-state drive.

The LiDAR sensor module uses RoboSense’s RS-16 LiDAR. The sensor has 16 LiDAR beams for distance measurements, with an accuracy of ± 2 cm, a vertical viewing angle of ± 15°, a measuring distance of up to 150 m, and an angular resolution ranging from 0.09° to 0.36°, and the working frequency of the module was 10 Hz. The IMU is an Xsens series MTi-300 inertial navigation unit which outputs high-precision pitch, roll, and yaw angles at a high frequency (400 Hz) (0.2°, 0.3°, and 1°) and has strong seismic and anti-interference properties.

The GNSS module was used as a module for testing the accuracy of the odometry pose of the mobile scanning system, which selects Real time kinematic (RTK) GNSS equipment from CHCNAV. The system has two satellite receiving antennas. RTK technology can be used in network mode to obtain high-precision positioning and orientation information. The positioning accuracy provided by the module was less than 1.5 cm, and the orientation accuracy was less than 0.09°, while working frequency was 10–50 Hz.

### 2.2. Software Framework

The system was divided into three modules: a high-precision orchard point cloud map building module, a canopy parameter module based on the point cloud model, and a test module based on manual measurement values and GNSS. The system was developed and deployed under the framework of the open-source Robot Operating System (ROS) [17], the development language is C++, and the nonlinear optimization library is Ceres Solver. In addition, to better visualize the point cloud processing effect, OpenGL was used to render and display the processing result. The system framework is shown in Figure 2.

### 2.3. Orchard Point Cloud Map Establishments

#### 2.3.1. Feature Point Extraction

Since the LiDAR sensor was in motion during the point cloud acquisition process, it was inevitable that there will be point cloud motion distortion. To correct the motion distortion, we first assumed that the LiDAR motion is a linear motion model and then used the pose obtained by the IMU and the LiDAR operating frequency to linearly interpolate the obtained point cloud, thereby obtaining the posture of the point cloud at any time and completing the motion distortion correction. The point cloud pose $T_{k}^{L}$ at time k is as follows:(1)$$T_{k}^{L}=\frac{t_{k}-t_{i}}{\Delta t}T_{\left(i,j\right)}^{L}$$
where $T_{\left(i,j\right)}^{L}$ is the transformation corresponding to a complete point cloud frame obtained from the IMU and Δt is the time when LiDAR receives a complete point cloud.

To improve the efficiency of subsequent point cloud matching, we used a method similar to that selected by Lego-LOAM [18] to extract stable feature points; the process is shown in Figure 3:

First, point cloud acquired by LiDAR was projected into a range image (the image resolution was 16 × 1800) and the distance r from the pixel value in the image to the sensor was calculated. Then, the ground points were evaluated according to the image distance value r, and the non-ground points were clustered using the Breadth-First Search (BFS) algorithm [19] to eliminate less than a certain number of cluster blocks. Finally, the curvature values of the ground point and the remaining non-ground points were calculated. The curvature calculation formula is as follows:(2)$$c=\frac{1}{\left|S\right|\cdot \left\|r_{i}\right\|}\left\|\sum_{j\in S,j\neq i}\left(r_{j}-r_{i}\right)\right\|$$
where S is the set of consecutive point clouds and the point clouds are located in the same row and on both sides of the calculated points.

In the non-ground point cloud, when the curvature of the point cloud was greater than the threshold value, it was selected as the edge feature point, and in the ground point cloud, when the curvature was greater than the threshold value, it was selected as the edge feature point and the point less than the threshold value was selected as the plane feature point. In order to ensure that the system can extract feature points uniformly [20], the image was divided into 6 sub-images according to angle. In each sub-image, 2 edge feature points with maximum curvature, 20 common edge points, 4 plane points with minimum curvature, and 40 common plane points were selected.

#### 2.3.2. LiDAR-IMU Odometry Based on ESKF

This module uses the pose of IMU and feature points to estimate the relative transformations of the consecutive LiDAR frames. This paper uses an iterative Kalman filtering method to transfer errors and to acquire odometry information.Define the state transition $x_{b_{k+1}}^{b_{k}}$ of IMU between time-step k and k + 1 with the error δx generated during the transition:(3)$$x_{b_{k+1}}^{b_{k}}=[p_{k+1}^{k},v_{k+1}^{k},q_{k+1}^{k},b_{a},b_{g},g^{k}]$$
(4)$$\delta x=[\delta p,\delta v,\delta \theta ,\delta b_{a},\delta b_{g},\delta g]$$
where $b_{k}$ is the states of IMU; $p_{k+1}^{k},q_{k+1}^{k}$ represent the translation and rotational changes of IMU from time-step k to k + 1; $b_{a},b_{g}$ represent the acceleration and gravity bias; and $g^{k}$ represents the gravity vector.The continuous-time linear error model of IMU can be described by Equation (5) [15]:(5)$$\overset{•}{\delta x}(t)=F_{t}\cdot \delta x(t)+G_{t}w$$
where F_t_ is the error state transfer matrix, G_t_ is the noise Jacobi matrix, and w is the Gaussian noise vector; more details can be found in the literature [15].The error propagation model $\delta x_{t_{\tau }}$ and covariance $p_{t_{\tau }}$ can be obtained by discretizing Equation (5):(6)$$\delta x_{t_{\tau }}=(I+F_{t_{\tau }}\cdot \Delta t)\cdot \delta x_{t_{\tau -1}}$$
(7)$$p_{t_{\tau }}=(I+F_{t_{\tau }}\cdot {\Delta t)p}_{t_{\tau -1}}{(I+F_{t_{\tau }}\cdot \Delta t)}^{T}+{(G}_{t_{\tau }}\cdot {\Delta t)Q(G}_{t_{\tau }}\cdot \Delta t)$$
where $t_{\tau }$ and $t_{\tau -1}$ are the consecutive time-steps of IMU measurements, $\Delta t=t_{\tau }-t_{\tau -1}$, and Q expresses the covariance matrix of w.Since LiDAR and IMU were calibrated offline [15], the measured value of LiDAR can be used as the observation value of IMU state propagation to update the state.Based on the resolution of $x_{b_{k+1}}^{b_{k}}$ [21] and the iterative Kalman filter principle [22], the state correction problem of the error can be converted into an iterative optimization problem:(8)minxe‖δx‖(pk)−+‖f(xbk+1bk−⊕δx)‖(JkMkJkT),  xbk+1bk−⊕δx=xbk+1bk=[pk+1k−+δpvk+1k−+δvqk+1k−⊗exp(δθ)ba−+δbabg−+δbggk−+δg]
where ‖·‖ is the Mahalanobis normal, J_k_ is the Jacobin of f(·) w.r.t. the measurement noise, M_k_ is the covariance matrix of measurement noise, f(·) is a stacked residual vector calculated from point-edge or point-plane pairs [18], ⊗ denotes the quaternion product, exp denotes the angle vector to quaternion rotation, and xbk+1bk− is the prior state.The iterative matching between adjacent feature points acquired by LiDAR can optimize Equation (8), and the Jacobian $H_{k,j}^{}$ (f(·) w.r.t. $\delta x_{j}$), the Kalman gain equation $K_{k,j}$, and the new error state $\delta x_{j+1}$ were calculated according to the Kalman filtering (KF) principle.
(9)$$K_{k,j}=P_{k}H_{k,j}^{T}{(H}_{k,j}P_{k}H_{k,j}^{T}+J_{k,j}M_{k}J_{k,j}^{T})$$
(10)δxj+1=δxj+Kk,j(Hk,jδxj−f(xbk+1bk−⊕δx))When iterative matching was completed, the final pose $x_{b_{k+1}}^{b_{k}}$ could be obtained from xbk+1bk−⊕δx, and the covariance matrix P_K+1_ required for state estimation at moment k + 1 could be obtained by Equation (11):(11)$$P_{k+1}=(I-K_{k,n}H_{k,n})P_{k}{\left(I-K_{k,n}H_{k,n}\right)}^{T}+K_{k,n}M_{k}K_{k,n}^{T}$$Synthesize the state in the global coordinate system [16] and proceed to the next step: optimization iteration after initialization.

#### 2.3.3. Mapping Module

The original version of the R-LINS algorithm uses the map-building module of the Lego-LOAM algorithm for global map construction, which results in the *z*-axis of the map drifting more severely during long-term operation. To overcome this situation, this paper introduces the rotation constraint method to correct the point cloud map.

Given that there was higher uncertainty along the *z*-axis and that the other two Degrees of Freedom (DoFs) were closer to the true property, we constrained the cost function by changing the orientation-based Jacobin matrix for optimizing the residual function according to [21]. To ensure that the map is always aligned with gravity, the formula for the modified Jacobin matrix based on Quaternary is shown below:(12)$$J_{\theta }^{z}=J_{\theta }^{}\cdot {\left(\hat{R}\right)}^{T}\cdot \;\Omega_{z},\Omega_{z}=\left[\begin{array}{ccc} \varepsilon_{x} & 0 & 0\\ 0 & \varepsilon_{y} & 0\\ 0 & 0 & 1 \end{array}\right]$$
where $J_{\theta }^{z}$ is the modified orientation-based Jacobin matrix, $J_{\theta }$ is the original orientation based Jacobin matrix, ${\left(\hat{R}\right)}^{T}$ denotes the estimation of the state in the last iteration, $\Omega_{z}$ is an approximation of the information matrix of the orientation w.r.t. the world coordinate system, and $\varepsilon_{x}$ and $\varepsilon_{y}$ can be obtained by the information ratio of the x- and y-axes orientation to the *z*-axis orientation.

The purpose of map building is to register the feature points into the global coordinate system and to build the loss function based on the relative LiDAR measurements and the final transformation matrix.
(13)$$C_{m}={\sum_{m}\left\|\omega^{T}(Rx+p)+d\right\|}^{2}$$
where C_m_ is the loss function to be optimized, R and p are the rotation and translation matrix corresponding to the final conversion matrix, m is the relative LiDAR measure, and ω and d are the normal vector of the plane and the distance from the plane, respectively. For a detailed introduction, see [14].

The Ceres nonlinear optimization library [23] was used to construct a Gauss–Newton algorithm to optimize the loss function to register the feature points in the global map. To better display the point cloud, OpenGL was used to visualize the point cloud. The final point cloud environment map was obtained as Figure 4:

#### 2.3.4. Ground Point Cloud Removal

After processing the noise, the ground point cloud needs to be removed. Since the ground in a standardized orchard is relatively flat and a point cloud map with normal elevation is obtained with the help of the rotation constraint method, ground point clouds can be rejected according to the location of LiDAR installation.

Assuming that the height of the sensor centre from the ground is d, the point clouds can be identified with z coordinates less than −z + 20 cm as ground point clouds for filtering.

### 2.4. Point Cloud Segmentation of Fruit Trees

After completing orchard environment map reconstruction and ground point cloud removal, the point cloud segmentation operation was then implemented to obtain the point cloud information of a single fruit tree. This system designed a two-step point cloud segmentation method to split the connected fruit trees. First, the point cloud was initially segmented according to the oversimplified European clustering algorithm and then determined the connected fruit tree point clouds that needed to be segmented twice according to the size of the point cloud. Then, Statistical Outlier Removal (SOR) filtering and European clustering were used for the second segmentation. Finally, the k-nearest neighbours search algorithm [24] was used to recover the filtered point cloud information.

#### 2.4.1. Point Cloud Clustering of Fruit Trees Based on Distance

The Euclidean clustering algorithm was used for point cloud clustering segmentation. The Euclidean clustering algorithm is a segmentation algorithm based on the distance between point clouds. It relies on the k-d tree for point cloud searches and uses the Euclidean distance between points as the basis for clustering. The process of the Euclidean clustering segmentation algorithm is as follows: (i) Find the point p_i_ in the point cloud, search for the nearest point to p_i_ using the k-d tree, and judge the distance between the surrounding points and p_i_. Store the points that are smaller than the threshold r into the sequence C. (ii) Find another point p_j_ in C and perform step 1 operations to update C. (iii) If the point in C is not updated, clear C and mark the point in C as the same type of object. (iv) Find a point outside C to perform the above operations until no new points are updated, and then complete the segmentation of all point clouds.

#### 2.4.2. Secondary Point Cloud Segmentation Based on Filtering Algorithm

After the initial segmentation, the unconnected trees were segmented, but a portion of the trees with their crowns joined together may be classified as the same fruit tree, which has implications for completing the canopy estimation of the trees. The distribution of point clouds where two canopies connect was noisy and scattered, and the variance of the distance between point clouds was higher than the other parts, so the point clouds can be filtered with statistical variance filtering. This method first calculated the distance value D between the point cloud and its neighbours and obtained the parameters of standard Gaussian distribution d~N (μ and σ) according to the distance value [25]:(14)$$\{\begin{array}{c} \mu =\frac{1}{nk}\sum_{i=1}^{k}\sum_{j=1}^{k}\sum_{l=1}^{k}D_{ijl}\\ \sigma =\sqrt{\frac{1}{\mathrm{nk}}\sum_{i=1}^{k}\sum_{j=1}^{k}\sum_{l=1}^{k}{\left(D_{ijl}-\mu \right)}^{2}} \end{array}$$
where k is the number of neighbouring point clouds and n is the total number of point clouds.

Finally, the average distance between each point and a neighbouring point was calculated, and the point was deleted if the average distance d was in a special interval determined by a Gaussian distribution. The expression for this interval Q is as follows.
(15)Q>μ+mσ or Q<μ−mσ
where μ and σ are the mean and standard deviation, respectively, and m is the threshold value indicating the standard deviation multiplier.

After SOR filtering, a lot of point clouds were lost, which seriously affects the morphological measurement of fruit trees. In order to recover the filtered point cloud, we used the distance query method to classify the filtered point cloud. First, set the filtered point cloud as the source point cloud and the corresponding cluster point cloud as the target point cloud. Use the source point cloud k nearest neighbour search algorithm to search for the nearest point on the target point cloud, and determine the attribution of the filtered point according to the category of the searched nearest point. The complete point cloud processing process is shown in Figure 5.

### 2.5. Extraction of Fruit Tree Canopy Parameters

#### 2.5.1. Crown Height Extraction

In this paper, the tree height measurement was considered the crown height because the trunks of the young pear trees were less exposed. The rough ground segmentation algorithm based on LiDAR installation distance was used in the previous study, which made it impossible to measure tree height accurately. The RANSAC [26] algorithm was used to fit the ground point cloud, and based on the obtained ground point cloud, the highest distance from the ground to the canopy point cloud was calculated to determine the canopy height. The ground-fitting effect and tree height measurements are shown in Figure 6a.

#### 2.5.2. Canopy Volume Calculation

Canopy volume is an important parameter in tree crown morphology. To obtain a relatively accurate canopy volume, we used the alpha-shape model for canopy volume description. Alpha-shape is a method of surface fitting using discrete point clouds, and the algorithm mainly obtains the contour envelope by designing rollers of different radii to obtain possible combinations of concave polygons. Contour reconstruction was coarse when the radius factor alpha was too large; in contrast, if the radius was too small, canopy contour tearing occurred. Canopy reconstruction based on the alpha-shape algorithm is shown in Figure 6b.

Alpha-shape reconstruction of the canopy can be thought of as a Delaunay triangulation based on alpha weights [27], which decomposes the canopy into a collection of tetrahedral compositions by triangulation. The volume of a single tetrahedron is as follows:(16)$$V_{}=\frac{1}{3}A\cdot h$$
where A is the volume of the triangle at the bottom of the tetrahedron and h is the height.

The final canopy volume can be determined by the summation of all tetrahedron volumes inside the alpha-shape of the canopy calculated by 3D Delaunay triangulation.
(17)$$V_{\mathrm{canopy}}=\sum_{i=0}^{n}V_{i}$$
where V_canopy_ is the canopy volume and n is the number of tetrahedrons.

## 3. System Performance Testing and Analysis

The experimental site was located in Meiwu Village, Fangzi District, Wei fang City, Shandong Province, and the experiment time was August 2020. The orchard was a modern standard orchard suitable for navigating with large-sized agricultural machinery and robot. The variety of peach trees planted in the orchard were Beijing No. 8 peach trees, which were 5–6 years old, with a row spacing of 5 m and a distance of 4 m. There were a small number of dead or small trees in the orchard, and the measurement area was about 200 m × 50 m, with 7 rows of fruit trees. The canopies in the orchard were not uniform in size, and there were a certain number of canopies connected together.

### 3.1. Comparison of Orchard Environment Reconstruction

To compare the point cloud-based reconstruction of the orchard environment, we used the data logging function in the ROS framework to store the data collected by LiDAR and the IMU and then used the data several times to carry out a comparison of different methods. The comparison experiment mainly focused on the drift of the point cloud environment map in the *z*-axis direction. The side view of the point cloud map built by the original method and the improved method is shown in Figure 7.

It can be seen in the Figure 7 that the point cloud map obtained by the improved algorithm has no significant drift in the z-axis direction (Figure 7a), while the point cloud map obtained by the original algorithm after a long period of operation has a large drift in the z-axis direction (Figure 7b).

In order to better analyse and compare the mapping effect, the absolute position information obtained by the GNSS carried by the robot was compared with the odometer information obtained by the system. In addition, the original system also participated in comparative experiments in order to verify the optimization results of the system. The mean and maximum values of the absolute value errors in the x, y, z directions were extracted as the evaluation indexes, and the results are shown in Table 1.

From Table 1, it can be found that the odometry information collected in this paper obtains a decimetre-level error compared with GNSS, and the maximum and average errors in several directions do not exceed 0.294 m and 0.162 m. Compared with the original version of the system, this paper makes significant progress in the direction of suppressing map *z*-axis drift, and the maximum and average errors are reduced by 1.249 m and 0.944 m, respectively. Due to the large computational requirements of the constraint algorithm, the running time of the system is slightly increased compared with the original algorithm, but this state does not have a significant impact on the system because the task of the system does not require high real-time performance.

### 3.2. Orchard Fruit Tree Segmentation Test

Fruit tree segmentation is also an important task in this paper. To ensure the accuracy of fruit tree segmentation, the choice of coefficients is very important.

Experiments changing the parameter m related to the distance interval of the SOR filter obtained the optimal parameter for segmentation of adjacent fruit trees in the orchard. Here, we used 8 different sets of m values to verify the results and set the proximity point of SOR to 50 and the classification threshold of European clustering to 20 mm. The test results are shown in Table 2.

Through comparison experiments, it can be found that the segmentation success rate is the best when m is taken as 0.01, and when m is larger than 0.04, the point clouds that are connected together in the canopy cannot be filtered out and the segmentation success rate cannot be improved. If the value of m is less than 0.01, the canopy point cloud will be split into multiple parts and the segmentation success rate will be reduced, so 0.01 is chosen as the experimental parameter in this paper; since the original algorithm has a high drift in the *z*-axis when constructing the orchard map, as shown in Figure 7b, this can lead to ground information not being extracted effectively, thus making fruit tree segmentation impossible. The overall segmentation effect of the orchard is as Figure 8:

### 3.3. Tree Height and Crown Volume Measurement Experiment

To verify the accuracy of the obtained tree height and canopy volume parameters, manual measurements were used to calculate these parameters and compared with the automatically obtained parameters. R^2^ and RMSE were used as the evaluation criteria. The calculation process is as follows:(18)$$R^{2}=1-\frac{SS_{\mathrm{res}}}{SS_{\mathrm{tot}}}$$
(19)$$\mathrm{RMSE}=\sqrt{\frac{1}{m}\sum_{i=1}^{m}(y^{i}-{\hat{y}}^{i}{)}^{2}}$$
where SS_res_ is the residual sum of squares, SS_tot_ is the total sum of squares, $\hat{y}$ is the true value, y is the predicted value, and m is the total number of samples.

In order to obtain the best alpha values for the canopy reconstruction parameters, a number of fruit trees with regular contour characteristics in the orchard were selected for comparison experiments by comparing 8 different sets of alpha values, and the experiments were conducted with manual measurements as reference. Since, when the alpha value is less than 0.1, the canopy tearing occurs, leading to an inability to estimate its volume, the experimental parameter alpha in this paper is chosen from 0.2 to 0.75, and the results of the comparison experiments are shown as Figure 9.

It can be seen from the figure that, when alpha is taken as 0.25, the overall canopy volume error of the test sample reaches a desirable level relative to other test groups, so this value is chosen as the canopy volume reconstruction parameter in this paper.

Fifty fruit trees were selected as test samples in the orchard, their crown height and crown volume were measured manually using the manual method, and the results were compared with the system measurements as real values. The experimenter used a measuring rod to measure the highest point of the tree from the ground as the tree height, and to achieve a better result, the same sample was measured three times and averaged as the final crown height measurement. The comparison of the final canopy height measurements is shown in Figure 10a.

Equations (16) and (17) are used to calculate the volume of the crown. The shape of the canopy of young pear trees is mostly ellipsoidal and conical, and to ensure the accuracy of manual measurements, fruit trees with distinct canopy morphologies were selected for the selection of measurement samples. The main parameters of morphology were measured using a tape measure, and the canopy volume was calculated using Equation (20):(20)$$\{\begin{array}{c} V=\frac{\pi x^{2}y}{12}\;\;\;\;\mathrm{cone}\\ V=\frac{\pi x^{2}y}{6}\;\;\;\mathrm{ellipsoid} \end{array}$$
where x is the diameter of the crown and y is the height of the crown.

The R^2^ and RMSE values of the canopy volume obtained by the two measurement algorithms are shown in Figure 10b.

In summary, this system can complete environmental reconstruction of large orchards without relying on GNSS and van extract the information of fruit trees in the map to complete overall canopy parameter measurements, which is an improvement compared with previous studies on canopy extraction for a single or few fruit trees. A rotation constraint algorithm was added to the lidar_imu odometer to constrain the point cloud drifts in the *z*-axis aspect, and the average error of the drift in the *z*-axis was reduced by 0.944 m compared to the original algorithm in a standardized orchard. Due to the fact that the point cloud at the connection between fruit trees was sparse compared with other parts, SOR filtering fusion clustering and KNN algorithm were used to segment the fruit trees, and the success rate of segmentation in the experimental orchard was 94.5%. After obtaining information on a single fruit tree in the orchard, point cloud statistics and alpha-shape reconstruction algorithms were used to calculate the tree height and crown volume. The R^2^ and RMSE values obtained from manual measurement of crown height were 0.93682 and 0.04337, respectively, and the corresponding values of crown volume were 0.8406 and 1.57308, respectively.

## 4. Conclusions

This paper develops a canopy mobile scanning system, which can be deployed in a tracked orchard robot and completes point cloud-based orchard environment reconstruction and tree segmentation in a map while the robot operates, and extracts canopy information of fruit trees according to the segmentation results.

The point cloud map is constructed using the lidar_imu tightly coupled odometry and rotation constraint algorithm, which is able to reconstruct the environment in an orchard environment with few feature points, and the maximum errors in the x, y, and z directions are no more than 0.198 m, 0.149 m, and 0.249 m in an orchard of size 200 m × 50 m. The point cloud segmentation method incorporating Euclidean clustering can complete segmentation of fruit trees in the orchard with a segmentation success rate higher than 94%, which creates conditions for subsequent measurement of the overall orchard canopy information.

This system uses the point cloud statistics and alpha-shape method to calculate the crown height and crown volume. The experiments show that the alpha-shape algorithm can better represent concave and convex surfaces of the canopy in an appropriate alpha parameter and obtained R^2^ and RMSE values of 0.8406 and 1.57308 compared with manual measurement.

## Figures and Tables

**Figure 1 sensors-21-00339-f001:**
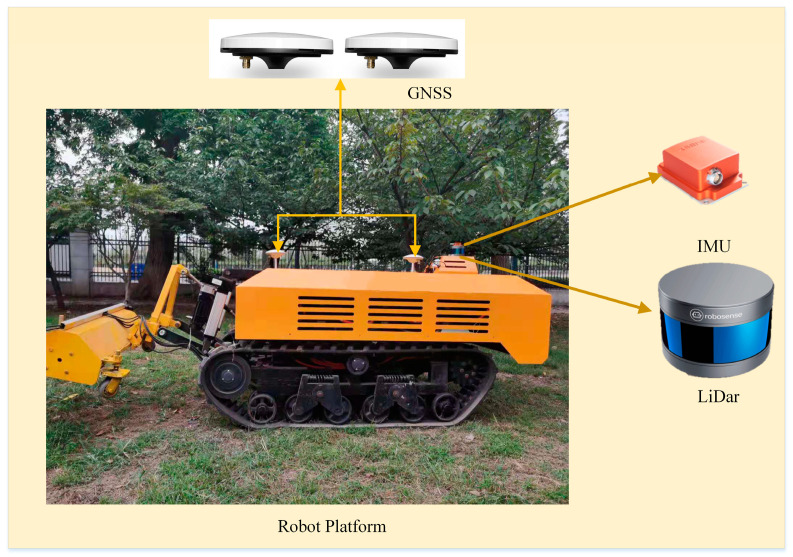
Tracked robot experimental platform and main sensors.

**Figure 2 sensors-21-00339-f002:**
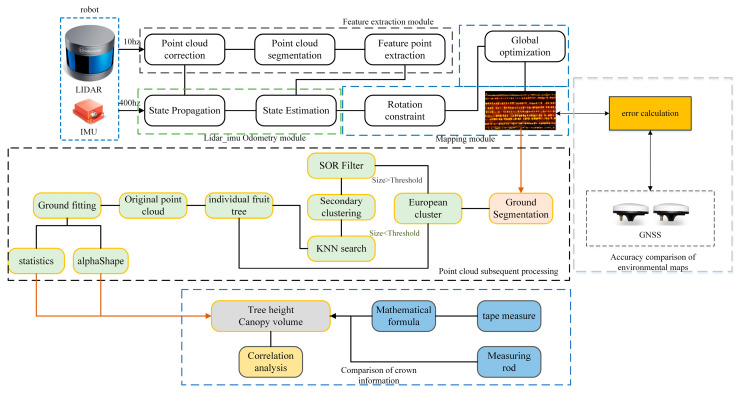
The framework of the software system: The top half of Figure 2 is the orchard environment reconstruction module, the middle part is the fruit tree canopy information measurement module, and the lower part and the right part are test modules.

**Figure 3 sensors-21-00339-f003:**
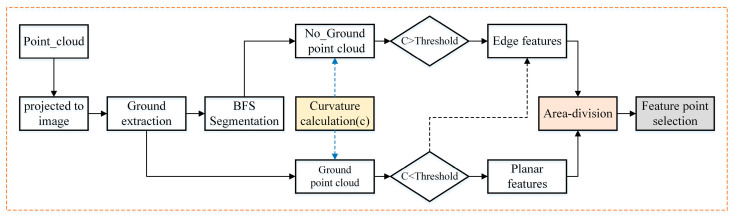
Feature point extraction process.

**Figure 4 sensors-21-00339-f004:**
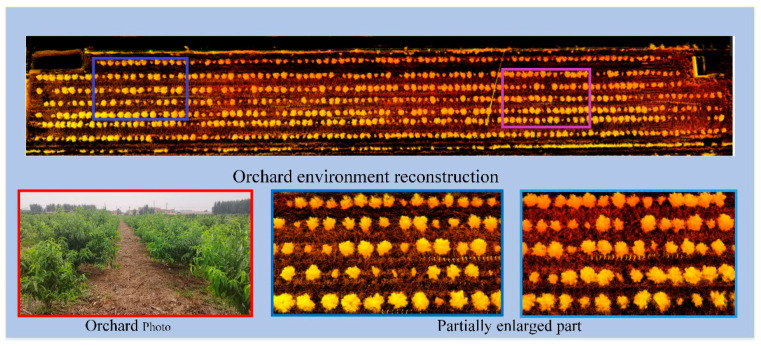
Natural environment photos and point cloud maps.

**Figure 5 sensors-21-00339-f005:**
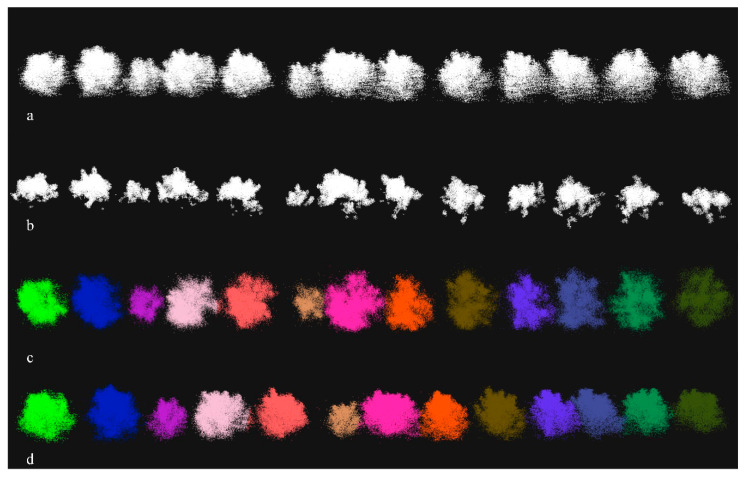
Fruit tree point cloud segmentation process: (**a**) the original point cloud, (**b**) the point cloud after Statistical Outlier Removal (SOR) filtering, (**c**) the point cloud after Euclidean clustering, and (**d**) the point cloud to restore point cloud information through a k-nearest neighbour (KNN) search algorithm.

**Figure 6 sensors-21-00339-f006:**
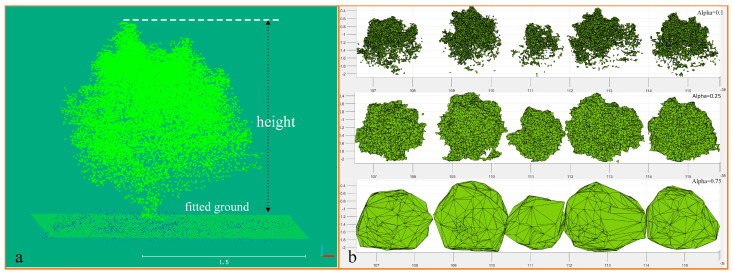
Tree height measurement (**a**) and canopy reconstruction effect under different alpha parameters (**b**).

**Figure 7 sensors-21-00339-f007:**
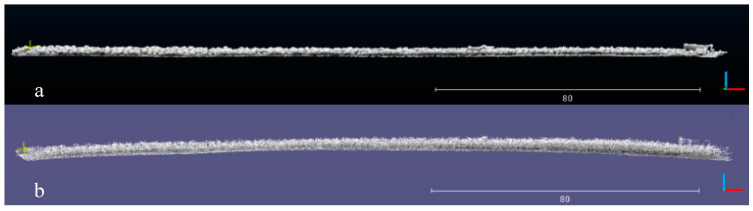
Side view of the orchard reconstruction environment based on point clouds: (**a**) improved and (**b**) unimproved mapping algorithms.

**Figure 8 sensors-21-00339-f008:**
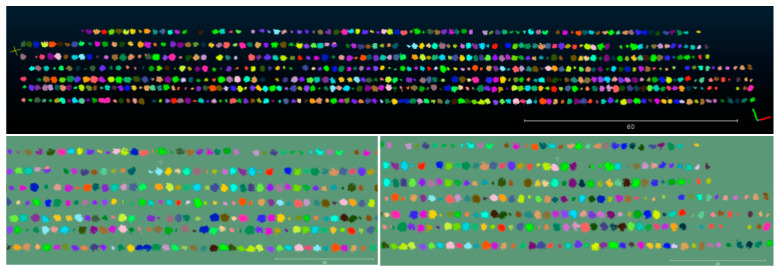
Point cloud segmentation effect: the upper part of Figure 8 is the overall segmentation view, and the lower part is the partially enlarged view.

**Figure 9 sensors-21-00339-f009:**
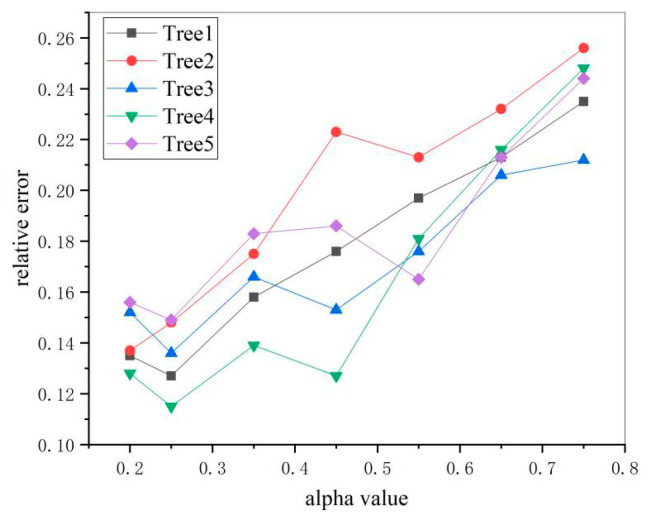
Canopy volume measurement results under different alpha values.

**Figure 10 sensors-21-00339-f010:**
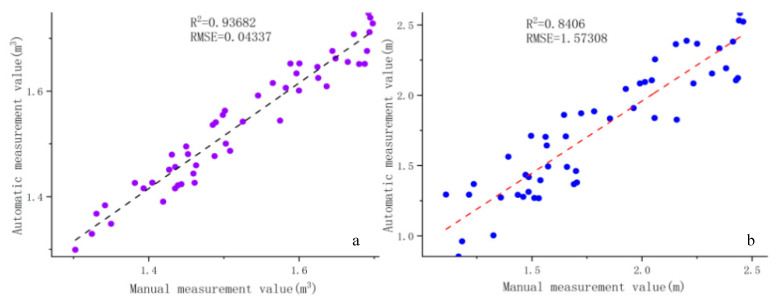
Comparison of manually measured values and automatically calculated values: (**a**) the comparison of tree height parameters and (**b**) the comparison of canopy volume.

**Table 1 sensors-21-00339-t001:** Comparison of errors in different directions.

Name	Max(x)	Max(y)	Max(z)	Mean(x)	Mean(y)	Mean(z)	Time
Original Method	0.201 m	0.189 m	1.816 m	0.095 m	0.106 m	1.106 m	1252.4 s
Our Method	0.198 m	0.194 m	0.249 m	0.098 m	0.118 m	0.162 m	1343.5 s

**Table 2 sensors-21-00339-t002:** Segmentation success rate under different parameters.

Number	Parameter	Successes	Fail	Success Rate
1	m = 0.2	487	118	80.4%
2	m = 0.1	487	118	80.4%
3	m = 0.08	487	118	80.4%
4	m = 0.04	485	120	80.1%
5	m = 0.01	572	33	94.5%
6	m = 0.008	504	51	83.4%
7	m = 0.006	461	144	76.1%
8	m = 0.004	368	237	60.8%

## Data Availability

The raw/processed data required to reproduce these findings cannot be shared at this time as the data also forms part of an ongoing study.

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
