# Peer review of "Mobile LiDAR Scanning System Combined with Canopy Morphology Extracting Methods for Tree Crown Parameters Evaluation in Orchards"

_sensors, 2021, doi:10.3390/s21020339_

Round 1
Reviewer 1 Report
The paper described the designed LiDAR scanning system based on 3D-SLAM algorithm. The extracting method is simple, but it can be widely used in orchard measurement.
There are some serious problems:
- The tile is “Extracting canopy morphology based on a mobile LiDAR scanning system in an orchard”. I suppose you will give a talking about canopy morphology extracting method, but the Section 2 mainly tells the designed LiDAR scanning system for extracting canopy morphology. There is no innovation in the extraction method of the paper.
- Figure 1: the Yellow arrow from the picture to the sensor? There are two GNSS antennas, but which are not described.
- Section 2.3.1: The description of feature point extraction is too simple for users to implement.
- Section 2.3.2: How the LiDAR-IMU tightly coupled odometry algorithm work? It is not clear.
- Section 2.3.3: Ground Point Cloud Removal can only be used on very flat ground. In fact, there are many previous methods that can be applied.
- The accuracy index or other parameters of the new LiDAR scanning system are not tested and specified.
- There is little information about the test site. The proposed canopy morphology extracting method may be mainly applicable for simple scene.
- Conclusions are too short. The applicability of LiDAR scanning system and canopy morphology extracting method should be analyzed.
The following defects are obvious.
- Line 23-24: “will have important application value” may be ““would have important application”?
- Line 112: “LiDAR” may be “the LiDAR sensor”
- Line 117, Line 124: “TL k”? “XL (k, i)”? Please check the symbol of the formula carefully.
- Line 132: the prior state and the error state is the same?
- Line 234-236: “5-a”, “5-b”, “5-c”, “5-d” the sub-figure numbers need to be standardized.
Reviewer 2 Report
In this paper, an orchard mapping and canopy measurement system based on the lightweight LiDAR-IMU odometry algorithm was develop. The system is deployed on a self-developed large-scale crawler robot, which can obtain global point cloud information of a large-scale orchard and segment and extract the fruit trees in the orchard through a point cloud segmentation algorithm. After complete segmentation, the robot can measure the tree height and canopy volume based on point cloud statistics and alpha-shape contour reconstruction methods
The paper is well written and straightforward. The following comments must be improved and clarified.
Mayor comments:
1) Line 295. “After many experiments, when the α and…”. How many experiments are you talking about?. Give data. Has this parameter to be recalculated for new application cases depending on the scanning distances, the type of trees, size trees, distance between the trees, etc?. Clarify this point. If it has to be recalculated then the methodology has a weak point here.
2) Line 320-321. “After several groups of tests, it can be seen that when the alpha is 0.25…”. How many test are you talking about? give data. Have these parameters to be recalculated for new application cases depending on the scanning distances, the type of trees, size trees, distance between the trees, etc?. Clarify this point. If it has to be recalculated then the methodology has a weak point here.
3) Line 334 -336. “The experimental results show that the environmental reconstruction algorithm in this paper is significantly better than the original algorithm in suppressing Z-axis drift, and the success rate of fruit tree segmentation can reach 94.5%.”. Which is the rate when the original algorithm is used?. What are the improvements of this new algorithm compared to the previous one?
4) The authors should give the computation time used by the environmental reconstruction algorithm for the case study and the characteristics of the hardware used.
5) The authors should give the comparison of computation times with the original environmental reconstruction algorithm.
6) Give an end summary paragraph (before of conclusions) highlighting the novelties and advantages of the proposed method and algorithm with respect to the existing ones.
Minor comments:
-There are two sections 2.3.2
-Section 2.5 is missing
-Figure 4 is not mentioned in the text
-I think that is better to use third person in a scientific manuscript
Round 2
Reviewer 1 Report
- Figure 1: the Yellow arrow from the picture to the sensor is still wrong!
- Line 97: There are two GNSS antennas, but which are not applied and described in the framwork.
- Line 129: “lidar” should be replaced by “LiDAR”.
- Figure 3, Line 132: what is the BFS algorithm?
- Line 256: “Section 2” may be “Section 2.5”
- Line 303-317: The GNSS is not used in the LiDAR-IMU system, but how the absolute position captured?
Reviewer 2 Report
Questions were answered
Author Response
Thank you very much!